# Semantics-aware Test-time Adaptation for 3D Human Pose Estimation

**Qiuxia Lin** [1]   **Rongyu Chen** [1]   **Kerui Gu** [1]   **Angela Yao** [1]

## Abstract

This work highlights a semantics misalignment in 3D human pose estimation. For the task of test-time adaptation, the misalignment manifests as overly smoothed and unguided predictions. The smoothing settles predictions towards some average pose. Furthermore, when there are occlusions or truncations, the adaptation becomes fully unguided. To this end, we pioneer the integration of a semantics-aware motion prior for the test-time adaptation of 3D pose estimation. We leverage video understanding and a well-structured motion-text space to adapt the model motion prediction to adhere to video semantics during test time. Additionally, we incorporate a missing 2D pose completion based on the motion-text similarity. The pose completion strengthens the motion prior's guidance for occlusions and truncations. Our method significantly improves state-of-the-art 3D human pose estimation TTA techniques, with more than **12%** decrease in PA-MPJPE on 3DPW and 3DHP.

## 1. Introduction

3D human pose estimation from images and videos is applicable for many scenarios, including human-computer interactions (Zheng et al., 2023), robots (Gong et al., 2022), and digital human assets (Moon et al., 2024). It is a challenging task because 3D ground truth labels are hard to acquire for arbitrary visual data. Common methods are trained on MoCap datasets (Mahmood et al., 2019), which provide accurate 3D labels but are limited in pose and appearance. As such, these methods may easily fail on in-the-wild images during inference (Guan et al., 2021; Zhang et al., 2020; Weng et al., 2022; Nam et al., 2023).

This paper tackles the adaptation of 3D human pose estima-

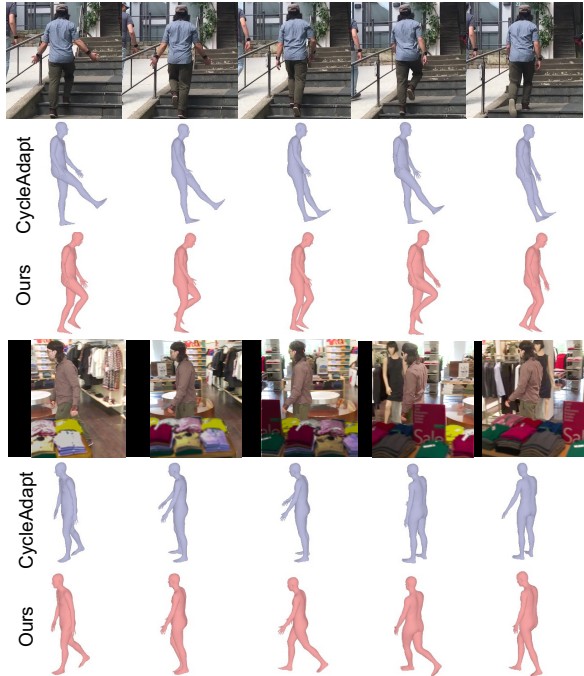

Figure 1: We provide two examples to illustrate the issues in TTA methods. Compared to CycleAdapt (Nam et al., 2023), our method enables accurate bending of the legs in the "climbing-the-stairs" scenario and "walking" in the occluded scenario, demonstrating the effectiveness of incorporating semantics-aware motion information for both visible and occluded keypoints.

tion models to in-the-wild videos. A practical alternative to improve predictions is Test-Time Adaptation (TTA). TTA directly uses the (unlabeled) target sequences to fine-tune the pre-trained model during inference. Often, pre-trained models exhibit a misalignment between their projected keypoints and 2D image evidence (Chen et al., 2025a). Based on this result, previous TTA methods (Guan et al., 2022; Nam et al., 2023) usually penalize the distance between the 2D-projections and 2D evidence provided by either ground truth or estimated 2D poses. However, such a scheme has two main challenges (Chen et al., 2023): 1) Depth ambiguity, which arises because numerous 3D pose solutions correspond to the same 2D pose. This ambiguity may improve 2D alignment but worsen 3D accuracy and plausibility.

[1]Department of Computer Science, National University of Singapore, Singapore. Correspondence to: Qiuxia Lin <qiuxia@comp.nus.edu.sg>.

*Proceedings of the 42nd International Conference on Machine Learning*, Vancouver, Canada. PMLR 267, 2025. Copyright 2025 by the author(s).

2) Missing 2D keypoints, *e.g.*, under occlusions and truncations. These keypoints have no guidance for improvement.

To alleviate the above issues, several works (Guan et al., 2022; Nam et al., 2023) leverage temporal information from the target videos. One strategy is to limit the prediction velocity, as a weak regularizer to narrow the feasible solution space (Guan et al., 2022). Another is motion denoising, which enforces a prior that predictions should remain close to the original or neighboring predictions (Nam et al., 2023). These improvements are rooted in the fact that denoising and smoothing the sequential predictions can filter out poor results. However, temporal smoothing may settle the sequence towards an average pose. Consider the upper panel of Fig. 1. The state-of-the-art CycleAdapt (Nam et al., 2023) aligns the 3D mesh to the visual 2D evidence, but results in the wrong pose where the legs are straight. With some semantic knowledge of the man's activity (climbing stairs), it is clear that his knees should be bent.

With this motivation in mind, we develop a semantics-aware motion prior to guide 3D predictions. We leverage vision-language models to describe the action of a given segment. This action text is projected with the predicted motion into a motion and text aligned representation space. Specifically, we use MotionCLIP (Tevet et al., 2022), with motion auto-encoder that aligns a motion manifold with the semantically-structured CLIP space (Radford et al., 2021). In our TTA, we align the estimated 3D pose sequence with the semantically identified action through a dedicated regularizer. In the "climbing-the-stairs" example, the predicted motion originally resembles "sliding", but after introducing our motion-text alignment, the resulting poses are adapted to have bent knees and match the true motion of stair climbing.

TTA can easily be influenced by challenging samples. For example, when there are occlusions, or truncation, there is no 2D evidence available for adaptation. Under such circumstances, the model lacks guidance on what to predict. The lower panel of Fig. 1 shows an example of a person walking, with the lower body occluded for the entire segment. CycleAdapt predicts a motion with a static lower body. In such cases, where 2D evidence is fully lacking, we observe that previous alignment in the motion-text space alone is insufficient for adaptation. This motivates us to strengthen the supervision by storing 2D predictions with high quality as exemplars and regard them as 2D pseudo labels in the subsequent optimization. Specifically, for the keypoints not provided with 2D evidence, we examine the similarity between the predicted motion and its text label in the feature space; if it exceeds a certain threshold, these keypoints will be completed by the 2D projection of model predictions. With 2D missing keypoints filled by text-aligned motion, we can offer stronger supervision on occluded or truncated body parts.

At the heart of our method is a semantics-aware motion prior that supports test-time adaptation for 3D human pose estimation. The prior effectively complements 2D projected supervision and remedies over-smoothing. When 2D keypoints are available, motion-text alignment greatly reduces the ambiguous 3D solution space; when they are unavailable, text-aligned motion prediction provides useful 2D projected poses to complete the missing information. In summary, we

- Highlight the interesting and significant problem of motion semantics, which existing TTA literature ignores or exacerbates;

- Propose a novel high-level motion semantics prior for TTA, by leveraging a motion-language model;

- Propose a text-aligned motion predictions to complete 2D poses in occlusion or truncation cases, significantly improving performance up to **12.5%** PA-MPJPE improvement on these challenging cases.

## 2. Related Work

### 2.1. Test-time Adaptation in 3D Pose Estimation

Generalizing 3D human pose models to in-the-wild data is challenging due to distribution shifts, driving interest in test-time adaptation methods with 2D information. Initially, the methods (Zhang et al., 2020; Guan et al., 2021; 2022; Weng et al., 2022) rely on ground truth 2D poses to rectify 3D predictions along with the proposed adaptation techniques. For instance, ISO (Zhang et al., 2020) leverages invariant geometric knowledge through self-supervised learning. BOA (Guan et al., 2021) introduces bilevel optimization to better integrate temporal information with 2D weak supervision. DynaBOA (Guan et al., 2022) extends BOA by using feature distance as a signal for domain shift with dynamic updates. DAPA (Weng et al., 2022) employs data augmentation by rendering estimated poses onto test images, guiding the model toward the target domain.

However, ground truth 2D poses are not always available in many real-world scenarios. Similar to our method, CycleAdapt (Nam et al., 2023) operates on estimated 2D data and introduces a cyclic framework, where the motion denoising module smooths 3D pseudo labels to mitigate the impact of noisy 2D inputs. Nonetheless, the inherent 2D-to-3D ambiguity still significantly degrades model performance, and unguided predictions often occur when 2D inputs are unreliable. Our method addresses this by leveraging explicit semantic understanding, *i.e.*, actions, to guide the model in predicting specific motion patterns. This is the first approach to emphasize high-level semantic information to

reduce the 2D-to-3D solution space in test-time 3D human pose estimation.

## 2.2. Human Motion Priors

Leveraging motion prior as supervision is commonly applied in video-based human pose estimation due to the lack of 3D annotations with RGB input. Trained on a large and diverse human motion database AMASS (Mahmood et al., 2019; Ionescu et al., 2013), the human prior provides a regularized motion space. Previous works mainly drag the predicted or generated motion to the regularized motion space to achieve the refinement (Kocabas et al., 2020; Shin et al., 2024; Rempe et al., 2021), infilling (Yuan et al., 2022; Zhang et al., 2021), denoising (Nam et al., 2023) goals. Specifically, VIBE (Kocabas et al., 2020) extends VAE (Pavlakos et al., 2019) and adversarial (Kanazawa et al., 2018) feasible prior from pose into motion space to improve temporal continuity of movement. GLAMR (Yuan et al., 2022) learns a VAE motion infiller to infill motion where the moving human is occluded. CycleAdapt utilizes the prior to denoise 3D meshes and improves human pose estimation during test-time with noisy 2D poses. However, those applications of human motion prior only optimize the predicted meshes of the segment to be valid motion without consideration of the motion semantics. Compared to other works, we are the first to split regularized motion space according to text labels extracted from the video segment to benefit motion prediction.

## 3. Preliminaries

### 3.1. Problem Setup

Given an arbitrary video, for each RGB frame $\mathbf{X}$, we aim to predict the 3D human poses in joint coordinates $\boldsymbol{J}^{3D} \in \mathbb{R}^{K \times 3}$ where $K$ is the number of keypoints. Following previous works, we utilize SMPL model (Loper et al., 2023) which takes as input the joint rotation $\boldsymbol{\theta} \in \mathbb{R}^{72}$ and shape $\boldsymbol{\beta} \in \mathbb{R}^{10}$ parameters along with a translation $\boldsymbol{t} \in \mathbb{R}^3$ w.r.t. the camera to produce 3D poses, formally

$$\boldsymbol{J}^{3D} = \text{SMPL}(\boldsymbol{\theta}, \boldsymbol{\beta}, \boldsymbol{t}). \qquad (1)$$

In our method, the predicted $\hat{\boldsymbol{\theta}}$, $\hat{\boldsymbol{\beta}}$ and $\hat{\boldsymbol{t}}$ are given by a human mesh recovery network (Kanazawa et al., 2018) which has been trained on the Human3.6M (Ionescu et al., 2013) dataset following the setting of (Guan et al., 2022; Zhang et al., 2020; Nam et al., 2023). We denote this network as our 3D human pose estimation model $f^{\text{HPE}}$.

During inference, the model is adapted with the help of 2D pose $\bar{\boldsymbol{J}}^{2D}$ from an off-the-shelf estimator (Cao et al., 2017) to improve the prediction of 3D poses. The predicted 2D poses $\hat{\boldsymbol{J}}^{2D}$ can be obtained by projection of predicted 3D poses $\hat{\boldsymbol{J}}^{3D}$:

$$\hat{\boldsymbol{J}}^{2D} = \Pi(\hat{\boldsymbol{J}}^{3D}) = \Pi(\text{SMPL}(\hat{\boldsymbol{\theta}}, \hat{\boldsymbol{\beta}}, \hat{\boldsymbol{t}})), \qquad (2)$$

where $\Pi$ is the camera projection function. Because our method does not involve direct operations on 3D poses, we will use $\boldsymbol{J}$ to denote 2D poses in the following descriptions.

### 3.2. Motion-language Model

A motion-language model integrates human motion and natural language modality. It utilizes pre-trained language models (Radford et al., 2018; Kenton & Toutanova, 2019), to incorporate rich semantic space for improved motion representation. This is achieved by training with motion-text data, which enables tasks like text-conditioned motion generation (Zhang et al., 2024; 2023) and motion captioning (Guo et al., 2022; Jiang et al., 2023). Our focus is on leveraging motion-language model to guide motion semantics for unlabeled test videos in TTA. We choose MotionCLIP (Tevet et al., 2022) for its text-aligned motion representation, which is optimally suited to our task.

MotionCLIP maps human motion sequences onto the CLIP space (Radford et al., 2021). It positions semantically similar motions closer together while maintaining clearer separation for distinct motions in the manifold. The motion-text affinity is measured using cosine similarity between the text CLIP feature from $f^{\text{CLIP}}$ and the motion feature from the motion encoder $f^{\text{MOTION}}$:

$$\text{sim}(\mathbf{s}, \mathbf{m}) = \cos\left(f^{\text{CLIP}}(\mathbf{s}), f^{\text{MOTION}}(\mathbf{m})\right). \qquad (3)$$

The motion-text pairs $(\mathbf{s}, \mathbf{m})$ are obtained from labeled motion dataset (Punnakkal et al., 2021), where $\mathbf{s}$ is text label and $\mathbf{m} = \{\boldsymbol{p}_1, \cdots, \boldsymbol{p}_T\}$ is motion segment of $T = 60$ frames. Each pose $\boldsymbol{p}_i \in \mathbb{R}^{24 \times 6}$ is represented in 6D format (Shi et al., 2019), including global and local rotations.

## 4. Method

Fig. 2 shows the overall framework. It consists of semantics-aware motion prior (left panel) and 2D pose update module (right panel). Using semantics-aware motion prior (Sec. 4.1), predicted motion and text label are aligned in a shared space of a motion-language model, supplementing other losses to enhance semantics awareness. In the 2D pose update module (Sec. 4.2), 2D poses are updated by EMA (right top panel) while the missing 2D poses are filled with text-aligned motion prediction (right bottom panel). This module strengthens the semantics-aware motion prior on occlusion. Details of each module are introduced as follows.

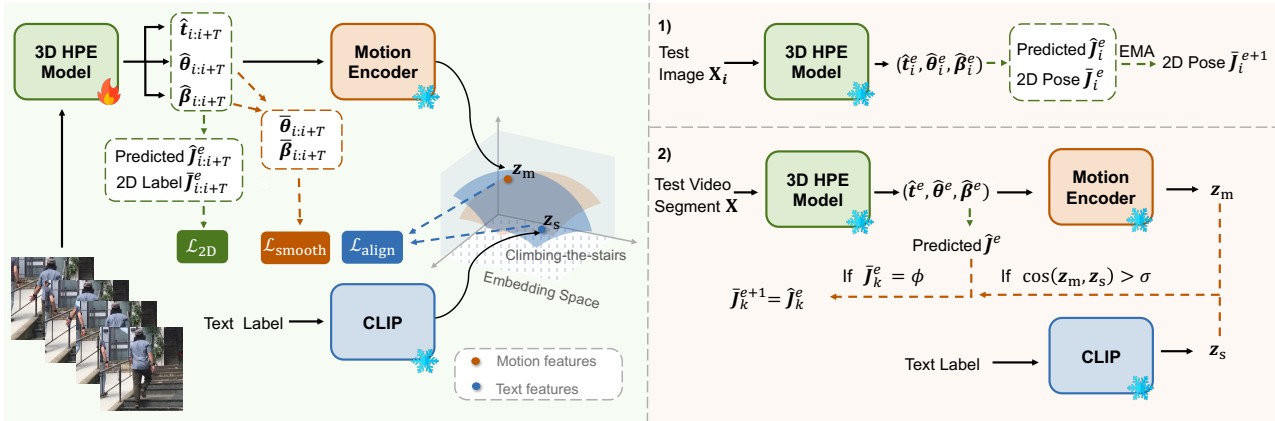

Figure 2: Overall framework: we perform model fine-tuning and 2D pose update in each epoch. Model Fine-tuning (Left): The extracted video segment and its text label will be used to update $f^{\text{HPE}}$ with 2D projection, alignment, and smooth losses. 2D Pose Update (Right): After model fine-tuning in each epoch, all images in the current test video are processed through $f^{\text{HPE}}$ to obtain predicted 2D and update the 2D keypoints using EMA. Subsequently, all video segments are checked to fill in any missing 2D keypoints by verifying conditions of motion-text alignment.

## 4.1. Semantics-aware Motion Prior

Since the 2D-to-3D space is highly ambiguous, the model supervised by 2D projection loss still suffers from depth ambiguity. However, when video segments are available, contextual semantic information about human motion is evident. We therefore can leverage it to reduce the 2D-to-3D ambiguity by specifying a pose space with clear semantic meaning. So aside 2D projection and temporal losses, we also use motion-text-aligned representation space learned by MotionCLIP (Tevet et al., 2022) to ensure the predictions grounded on the video semantics. This is achieved by aligning the features of predicted motion and text labels of motion semantics in the shared embedding space.

**Text labeling.** We leverage a Vision-Language Model (VLM) (Achiam et al., 2023) for human motion description and matching with a motion dictionary defined in (Punnakkal et al., 2021). In general, the inputs contain prompts specifying requirements, a video segment, and motion dictionary. The outputs are text labels, like running, sitting with feet crossed, or walking upstairs, which can be assigned to each frame within the video segment. We provide some examples of used text labels in our method as follows:

```
walking
sitting with feet crossed
standing with knees bent
throw something with the right hand and walking
pose with bent leg and transition and walking
…
```

More details about the VLM prompts and label verification can be found in Appendix Sec. B.

**Video segment processing.** After the text label for motion

in each image is obtained, we aim to align the the projected motion and text features within the shared space of Motion-CLIP. To process video segments for training, we divide the video into segments and only keep those where all images share the same text label. This is to avoid misalignment caused by mixed motions as much as possible. To further improve alignment, we adopt weighted sampling and prioritize those segments whose motion predictions deviate from the true semantics. Specifically, the sampling weight is defined as 1 minus cosine similarity between predicted motion and its text label in MotionCLIP.

**Motion-text feature alignment.** Given a batch of video segments, we pass them through our 3D human pose estimation model $f^{\text{HPE}}$, to obtain pose motion predictions. For one single video segment $\mathbf{X}_{i:i+T}$, we first retrieve the exemplar motion segment in the motion dictionary (Punnakkal et al., 2021) based on the text label $\mathbf{s}_i$. Next, we change pose prediction $\hat{\boldsymbol{\theta}}_{i:i+T}$ outputted from $f^{\text{HPE}}$ to 6D format $\hat{\boldsymbol{p}}_{i:i+T}$ and replace the global rotation with that of the retrieved motion segment to obtain $\hat{\boldsymbol{p}}'_{i:i+T}$. This focuses on updating only the local poses that are more text-relevant while avoiding the optimization towards specific global orientations present in the motion training data (Punnakkal et al., 2021). As shown in Fig. 2 (left), the $\hat{\boldsymbol{p}}'_{i:i+T}$ is forwarded into the motion encoder $f^{\text{MOTION}}$, while text label $\mathbf{s}_i$ is forwarded into the CLIP model $f^{\text{CLIP}}$. Semantics-aware motion prior is applied by motion-text feature alignment on the shared embedding space:

$$\mathcal{L}_{\text{align}} = 1 - \text{sim}(\mathbf{s}_i, \hat{\boldsymbol{p}}'_{i:i+T}), \qquad (4)$$

where $\text{sim}(\cdot, \cdot)$ is introduced in Eq. 3.

The gradient is backpropagated to update the parameters of

the pose estimation model $f^{\mathrm{HPE}}$. Both the motion encoder and CLIP model are frozen in our framework so as not to affect the already structured feature space. With the motion feature better aligned with the text feature, the motion prediction can be performed based on a smaller semantic space, thus reducing the 2D-to-3D ambiguity.

### 4.2. 2D Pose Update

In our work, we leverage a 2D estimator (Cao et al., 2017) to provide 2D pose estimates $\bar{J}$ and its corresponding visibility $v$, typically setting a threshold to exclude uncertain estimates (Guan et al., 2022; Nam et al., 2023). Therefore, for the keypoint $k$ on the test frame, the projection loss can be applied with L1 distance as:

$$\mathcal{L}_{\mathrm{2D}} = \sum_{k} v_k \cdot ||\hat{J}_k - \bar{J}_k||_1, \ v_k = \mathbb{1}\{\bar{J}_k \neq \emptyset\}. \quad (5)$$

Note that the projection loss is held for each frame in the video segment and the frame subscript is omitted for clarity.

Clearly, the 2D projection loss mentioned above cannot provide supervision for missing 2D keypoints, specifically $v_k = 0$. However, the semantic meaning of human motions can be highly evident even under occlusion or truncation by inferring from the video context. Therefore, we want to utilize the semantics-enhanced motion prediction to suggest possible 2D positions for model training. To this end, we manage to fill in the missing 2D keypoints with text-aligned motion prediction if the motion feature has a high cosine similarity with its text feature:

$$\bar{J}_k^{e+1} = \hat{J}_k^e, \ \text{if } \bar{J}_k^e = \emptyset \text{ and } \mathrm{sim}(\mathbf{s}, \hat{\boldsymbol{p}}') > \sigma, \quad (6)$$

where $e$ subscript represents the epoch number. The $k$-th joint of the 2D prediction $\hat{J}_k^e$ is assigned to the estimated 2D $\bar{J}_k^{e+1}$ in the next epoch if the motion aligns with the text and the estimated keypoint is excluded, *i.e.*, $\bar{J}_k^e = \emptyset$.

However, the individually predicted 2D poses can still be noisy. The text-aligned motion predictions are based on each video segment and require model feedback to update temporally consistent predictions by incorporating all suggested 2D positions. To this end, we aim to update the estimated 2D alongside the predicted 2D keypoint, using an EMA (Exponential Moving Average) update with update factor $\alpha$, given by:

$$\bar{J}_k^{e+1} = \alpha * \bar{J}_k^e + (1 - \alpha) * \hat{J}_k^e. \quad (7)$$

The EMA update is performed at the end of each epoch, and the updated pose $\bar{J}_k^{e+1}$ will be used as supervision for prediction $\hat{J}_k^{e+1}$ following Eq. 5 in the next epoch $e + 1$. Here, we not only update the fill-in 2D but also the original estimated 2D, as both can boost the performance.

Finally, we can summarize the updates of all cases as:

$$\bar{J}_k^{e+1} = \begin{cases} \alpha * \bar{J}_k^e + (1 - \alpha) * \hat{J}_k^e, & \text{if } \bar{J}_k^e \neq \emptyset, \\ \hat{J}_k^e, & \text{if } \bar{J}_k^e = \emptyset \ \& \ \mathrm{sim} > \sigma, \\ \emptyset, & \text{if } \bar{J}_k^e = \emptyset \ \& \ \mathrm{sim} \leq \sigma. \end{cases} \quad (8)$$

Specifically, in epoch 1, we rely on the 2D pose estimates with uncertain values excluded. By the end of epoch 1, EMA update is applied to all estimates with predicted 2D, followed by a fill-in for some missing 2D estimates. This process is repeated at the end of each subsequent epoch. We provide an illustration of 2D pose update in Fig. 2 (right).

### 4.3. Overall Training

For each new video, we fine-tune the pre-trained model over several epochs for adaptation. The overall framework is shown in Fig. 2. The 2D pose updates (Eq. 8) and model fine-tuning are conducted separately, with the 2D poses remaining fixed during the fine-tuning process. And in each epoch, similar with (Nam et al., 2023), we have a smoothing loss with the motion denoise module:

$$\mathcal{L}_{\mathrm{smooth}} = ||\hat{\boldsymbol{\theta}} - \bar{\boldsymbol{\theta}}||_1 + ||\hat{\boldsymbol{\beta}} - \bar{\boldsymbol{\beta}}||_1, \quad (9)$$

where the $\bar{\boldsymbol{\theta}}$ comes from the motion denoise module, and $\bar{\boldsymbol{\beta}}$ is the averaged shape estimation with a sliding window operation. Together with motion-text feature alignment loss, 2D projection loss, and loss weight $\lambda$, our training objective in each epoch can be summarized as:

$$\mathcal{L}_{\mathrm{overall}} = \lambda_1 \mathcal{L}_{\mathrm{2D}} + \lambda_2 \mathcal{L}_{\mathrm{align}} + \mathcal{L}_{\mathrm{smooth}}. \quad (10)$$

## 5. Experiments

### 5.1. Setup

**Datasets.** We follow the adaptation tasks from previous work (Zhang et al., 2020; Nam et al., 2023), using **Human3.6M** (Ionescu et al., 2013) as the labeled training dataset and **3DPW** (Von Marcard et al., 2018) and **3DHP** (Mehta et al., 2017) as the unlabeled test datasets. Human3.6M is a widely used indoor dataset comprising 3.6 million images annotated with 2D and 3D labels. Despite its extensive pose diversity, the HMR network (Kanazawa et al., 2018) pre-trained on this dataset often struggles to generalize to in-the-wild settings due to its limited variation in appearance and environmental factors. The 3DPW test set comprises 37 videos and presents a significant challenge as it is an outdoor dataset featuring diverse scenarios, including subject occlusion and varied environments. The 3DHP test set includes 6 videos with both indoor and outdoor scenarios, and it presents an additional challenge by featuring some poses not seen in Human3.6M. Furthermore, we validate our method on a egocentric dataset EgoBody (Zhang et al., 2022), which presents severe body truncation with missing 2D evidence.

Table 1: Test-time adaptation comparison results on 3DPW (Von Marcard et al., 2018) and 3DHP (Mehta et al., 2017) based on HMR network pre-trained on Human3.6M (Ionescu et al., 2013). Our method achieves the best performance across all metrics on 3DPW (Von Marcard et al., 2018) and 3DHP (Mehta et al., 2017) datasets, surpassing state-of-the-art methods.

| Method | 3DPW | | | 3DHP | |
|---|---|---|---|---|---|
| | MPJPE↓ | PA-MPJPE↓ | MPVPE↓ | MPJPE↓ | PA-MPJPE↓ |
| Pre-trained HMR | 230.3 | 123.4 | 253.4 | 218.5 | 119.6 |
| BOA (Guan et al., 2021) | 137.6 | 76.2 | 171.8 | 135.3 | 88.5 |
| DynaBOA (Guan et al., 2022) | 135.1 | 73.0 | 168.2 | 130.7 | 81.8 |
| DAPA (Weng et al., 2022) | 108.0 | 67.5 | 129.8 | - | - |
| CycleAdapt (Nam et al., 2023) | 87.7 | 53.8 | 105.7 | 110.3 | 74.4 |
| Ours | **76.4** | **47.2** | **94.0** | **101.3** | **65.1** |

Table 2: Test-time adaptation comparison results on Ego-Body (Zhang et al., 2022) and 3DPW (Von Marcard et al., 2018) based on HMR2.0 network (Goel et al., 2023). Our method achieves better performance than CycleAdapt (Nam et al., 2023).

| Method | EgoBody | | 3DPW | |
|---|---|---|---|---|
| | MPJPE↓ | PA-MPJPE↓ | MPJPE↓ | PA-MPJPE↓ |
| HMR2.0 | 118.2 | 69.6 | 80.2 | 53.3 |
| CycleAdapt | 108.7 | 60.4 | 73.7 | 46.8 |
| Ours | **104.6** | **56.1** | **71.2** | **44.3** |

Table 3: Ablation study of method components on the 3DPW dataset (Von Marcard et al., 2018) based on HMR network, demonstrating the incremental improvements achieved by each component.

| Alignment | 2D EMA | 2D Fill-in | MPJPE↓ | PA-MPJPE↓ |
|---|---|---|---|---|
| ✗ | ✗ | ✗ | 87.7 | 53.8 |
| ✓ | ✗ | ✗ | 79.3 | 49.6 |
| ✓ | ✓ | ✗ | 78.0 | 48.7 |
| ✓ | ✓ | ✓ | **76.4** | **47.2** |

**Evaluation metrics.** We report three evaluation metrics: Mean Per Joint Position Error (MPJPE) measuring the Euclidean distance between the predicted joints and the ground truth joints with aligned root joint; Procrustes-Aligned MPJPE (PA-MPJPE) measuring MPJPE after aligning predictions with ground truth in 3D space; Mean Per Vertex Position Error (MPVPE) measuring the average error across all vertices. All results are reported in millimeters. The results on 3DPW contain all evaluation metrics. Since 3DHP does not provide ground truth SMPL parameters, we report only MPJPE and PA-MPJPE for this dataset.

### 5.2. Implementation Details

At the start of test-time adaptation for each test video, the model parameters are initialized with the pre-trained values, following (Nam et al., 2023). We employed the Adam optimizer (Kingma & Ba, 2014) with parameters set to $beta1$ = 0.5, $beta2$ = 0.9, and a learning rate of 5.0e-5. A cosine scheduler is used with a minimum learning rate of 1.0e-6. The input images are resized to 224×224, and the frame number of each video segment is 60. We use a batch size of 4 and the total training epoch is 6. The hyperparameters are $\lambda_1 = 0.1$, $\lambda_2 = 0.2$, $\sigma = 0.75$, $\alpha = 0.9$. We use Openpose (Cao et al., 2017) to provide 2D poses with a confidence threshold of 0.3 (Gu et al., 2024). The comparison methods include state-of-the-art test-time adaptation

methods: BOA (Guan et al., 2021), DynaBOA (Guan et al., 2022), DAPA (Weng et al., 2022) and CycleAdapt (Nam et al., 2023).

### 5.3. Quantitative Results

As shown in Tab. 1, the pre-trained HMR network performs poorly on the 3DPW dataset, highlighting the limited generalization ability of pre-training on low-variability scenarios. While previous methods (BOA (Guan et al., 2021), DynaBOA (Guan et al., 2022), DAPA (Weng et al., 2022)) show improvements, they heavily rely on ground truth 2D labels and are prone to failure with noisy 2D inputs. In contrast, CycleAdapt (Nam et al., 2023) employs cyclic adaptation between the motion denoise module and the HMR network, effectively mitigating the impact of noisy 2D poses. Our method outperforms CycleAdapt by **12.9%** in MPJPE and **12.3%** in PA-MPJPE, demonstrating the effectiveness of inducing semantics-aware motion prior to improve 3D predictions. Similar promising results are observed on 3DHP. Specifically, our method significantly surpasses others, achieving improvements of **8.2%** of MPJPE and **12.5%** of PA-MPJPE over CycleAdapt.

We further validate our method by upgrading the backbone to the state-of-the-art HMR2.0 (Goel et al., 2023), which is trained with multiple in-the-wild datasets, on the more challenging EgoBody dataset. As shown in Tab. 2, our method continues to outperform CycleAdapt with a stronger back-

Table 4: Analysis of semantics-incorporated strategies on the 3DPW dataset (Von Marcard et al., 2018) based on HMR network. Building on CycleAdapt (Nam et al., 2023), we design two methods to enhance pose semantics with generated motion sequences from (Jiang et al., 2023). "*downtown_rampAndStairs*" and "*downtown_bar*" highlight different challenges. The former involves depth ambiguity when viewing people ascending stairs from behind, while the latter includes obstacles in a low-lighting bar environment. Our method consistently improves across videos while others do not.

| Method | downtown_rampAndStairs | | downtown_bar | | All videos | |
|---|---|---|---|---|---|---|
| | MPJPE↓ | PA-MPJPE↓ | MPJPE↓ | PA-MPJPE↓ | MPJPE↓ | PA-MPJPE↓ |
| CycleAdapt (Nam et al., 2023) | 94.5 | 60.4 | 97.3 | 65.0 | 87.7 | 53.8 |
| +Motion discriminator (Kocabas et al., 2020) | 85.7 | 52.9 | 117.9 | 72.2 | 115.0 | 69.1 |
| +Unpaired local poses (Jiang et al., 2023) | 82.3 | 50.6 | 102.6 | 71.7 | 88.2 | 54.5 |
| **Ours** | **77.8** | **49.2** | **83.8** | **58.2** | **76.4** | **47.2** |

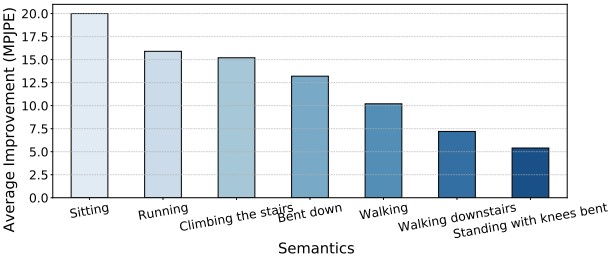

Figure 3: The improvement distribution on 3DPW dataset based on HMR network. We only show the top 7 semantics for brevity. The improved motions include both common and rare semantics found in action datasets.

bone. This indicates that the loss of semantic information still happens in the state-of-the-art backbone.

### 5.4. Analysis Experiments

**Ablations on the method components.** As shown in Tab. 3, we conduct an ablation study on 3DPW (Von Marcard et al., 2018) to evaluate the impact of each component in our method. Firstly, by adding semantics-aware motion prior, the performance can be improved from MPJPE 87.7mm to 79.3mm, which verifies the effectiveness of integrating semantic information in 3D predictions. We then introduce EMA and fill-in incrementally, progressively refining 2D poses and reinforcing the guidance of the motion prior. Each addition can further enhance model performance. The above experiments clearly validate the contribution of each component to the overall performance.

**Semantics-incorporated strategies analysis.** To the best of our knowledge, no existing works explicitly use semantic information to improve 3D motion in TTA. Therefore, we verify our method through comparison to the most relevant components from the literature, as shown in Tab. 4. Specifically, we use CycleAdapt (Nam et al., 2023) as a baseline, which lacks explicit semantic guidance. We then evaluate two alternative strategies for incorporating semantics,

replacing our proposed motion-text alignment with each: motion discriminator (Kocabas et al., 2020) (second row) and unpaired local poses (third row). Through adversarial loss or L1 minimization, the predicted motion is enforced to align with the generated motion from (Jiang et al., 2023) by sharing the same semantics as the video segment.

We implement the first comparison using motion discriminator. It guides motion prediction by distinguishing between predicted and generated motions of the same semantic meaning. As shown in Tab. 4, despite improvement in some cases, the overall performance is worse than CycleAdapt. One reason is the difficulty of stabilizing adversarial training; the other is that the discriminator, pre-trained to refine unrealistic motion rather than semantic differentiation.

We next implement the second comparison by using the local poses from the generated motion to supervise the predicted motion. Note that the local poses do not align with the video segment but share the same semantics. To our surprise, it shows effectiveness in some cases and achieves overall performance comparable to CycleAdapt. This may be because strong unpaired local pose supervision pulls predictions into a specific semantic space, while 2D evidence helps prevent misalignment. However, in the case of "*downtown_bar*", many 2D evidences are unavailable due to occlusion or poor lighting. Accordingly, the method predicts results identical to the generated motion, ignoring the actual motion in the test video.

Compared to the above two methods, our method avoids the need for additional motion data generation while simultaneously delivering consistent improvements.

**Improvement distribution.** In Fig. 3, we illustrate the average MPJPE improvement for the top 7 semantics. Our method enhances both common and less frequently observed motions in human action datasets, such as bending down and standing with bent knees.

**Runtime analysis.** We follow the same runtime evaluation setup as CycleAdapt (Nam et al., 2023) to ensure fair com-

Table 5: Runtime analysis (ms) on the 3DPW dataset. Our method introduces a modest runtime increase compared to CycleAdapt, but is still much faster than other TTA methods.

| Method | BOA | DynaBOA | DAPA | CycleAdapt | Ours |
|--------|-----|---------|------|-----------|------|
| Runtime | 840.3 | 1162.8 | 431.0 | **74.1** | 107.6 |

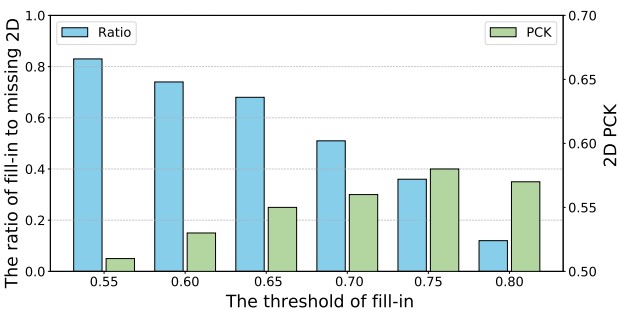

Figure 4: The relationship between the fill-in threshold, number of keypoints, and 2D keypoint PCK. Increasing the threshold leads to a decrease in the ratio of filled-in keypoints, but an improvement in their quality, as indicated by a higher PCK.

parison. As shown in Tab. 5, methods like BOA (Guan et al., 2021), DynaBOA (Guan et al., 2022), and DAPA (Weng et al., 2022) have high per-frame runtime due to complex optimization or extra synthesis rendering. Compared to CycleAdapt, our method introduces an additional 33.5ms runtime per frame, 85.4% of which comes from the VLM, and 14.6% from our proposed framework components. However, it is still much faster than other TTA methods.

**2D fill-in analysis.** We try different fill-in thresholds in Eq. 8 (from 0.55 to 0.80) to evaluate the quality and quantity of 2D keypoints. Fig. 4 shows as we increase the fill-in threshold, fewer new 2D keypoints are added (blue bars), but their PCK improves (green bars). Increasing the threshold helps use reliable but fewer predicted 2D keypoints in training, which have high feature similarity with their text labels.

Tab. 6 illustrates the overall performances on different thresholds. We find out that using 0.75 threshold adds only 36% more 2D keypoints, yet the high quality (PCK=0.58) helps achieve 76.4mm MPJPE. Even with a moderate quality (PCK=0.53), adequate new keypoints can still positively impact the model when the threshold is 0.60. Overall, the experiment demonstrates that balancing the quality and quantity of 2D keypoints can enhance model performance.

### 5.5. VLM Failure Cases

We manually examined 5,000 frames from the 3DPW dataset, where 96.4% is accurate, as the actions are rela-

Table 6: Comparison of fill-in threshold influence on model performance (MPJPE) on the 3DPW dataset. Balancing 2D keypoint quantity and quality can boost model performance.

| Threshold | 0.55 | 0.60 | 0.65 | 0.70 | 0.75 | 0.80 |
|-----------|------|------|------|------|------|------|
| MPJPE | 78.1 | 76.9 | 77.2 | 77.0 | **76.4** | 77.2 |

Table 7: Comparison on VLM failure cases: initial performance without adaptation, performance after adaptation with incorrect VLM label and correct (oracle) label, respectively. Adaptation with incorrect VLM label can still reduce MPJPE; accurate label yield greater improvement.

| Method | MPJPE |
|--------|-------|
| Initial | 213.8 |
| Incorrect VLM label | 92.2 |
| Correct (oracle) label | **83.7** |

tively simple and VLM is reliable. For the remaining 3.6%, the VLM hallucinates actions based on background, e.g. a person picking up foil is mistaken as "golfing" as the foil resembles a golf club, or "eating something using both hands" and "cast fishing pole" in scenes with fruit stands and rivers, respectively. The visualization of failure cases can be found in Appendix Fig. 11.

Despite these VLM failures, our method maintains robustness. As our semantics-aware alignment is a regularizer, its impact is limited if it conflicts with 2D projection loss. Tab. 7 shows that the adaptation with the wrong label still reduces the MPJPE, but not as much as if the correct (oracle) action label were applied.

### 5.6. Qualitative Results

We perform a qualitative comparison to visually assess the effectiveness of our method. In Fig. 5, we first illustrate that without 2D pose updates, the model predictions for the truncated part revert to average poses. Specifically, in epoch 2, frames highlighted with a green square show some walking poses. However, by epoch 4, these walking poses are back to standing poses (red squares). With 2D pose updates ("Ours"), more frames accurately reflect the intended action. This result verifies the necessity of using 2D pose update to preserve motion semantics.

Furthermore, we demonstrate that CycleAdapt (Nam et al., 2023) often predicts standing poses where 2D pose estimates are missing. Our method consistently estimates motions closer to the ground truth, verifying the feasibility of using semantic information to refine predicted motions. More visualizations can be found in Appendix Sec. C.

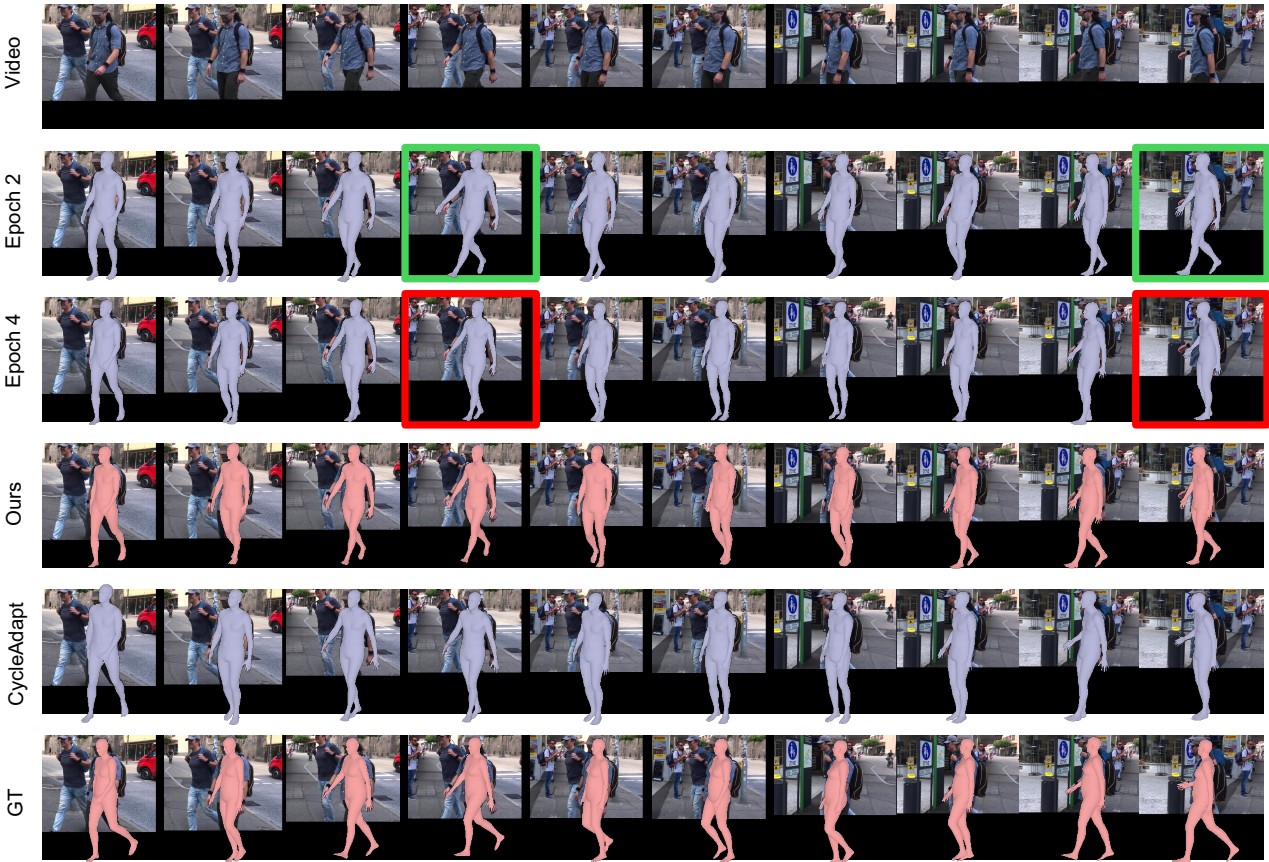

Figure 5: Qualitative comparison on one video sequence from 3DPW dataset (Von Marcard et al., 2018) with text label "Walking". From epoch 2 to epoch 4, results show that motion-text alignment alone is insufficient to preserve motion semantics with missing 2D data. We use green and red squares to highlight the transition from walking to standing poses. Incorporating 2D pose updates (Ours) helps mitigate this problem. Compared to CycleAdapt (Nam et al., 2023), which shows static standing foot under image truncation, ours captures the natural alternation of foot and knee bending during walking, aligning more closely with the actual walking sequences.

## 6. Conclusion and Limitations

We propose a novel text-guided test-time adaptation framework for solving 3D human pose estimation on unseen videos. Previous methods employ temporal smoothing to mitigate the impact of noisy 2D labels and depth ambiguity, but this often leads to average results and even stationary poses when 2D evidence is lacking. We address them by incorporating motion semantics from a well-trained motion-text space, refining 3D pose predictions in a correct semantic solution space. Moreover, the text-aligned motion predictions will be used to update 2D labels, enhancing the model understanding of the true motion. Our method ensures semantically consistent pose predictions across video sequences, outperforming state-of-the-art methods in both quantitative and qualitative evaluations.

However, our work has certain limitations: the refined motion under occlusion may not fully capture the actual pattern of the subject, with discrepancies in step frequency or duration; ambiguities like motion blur (Chen et al., 2025b) remains an open direction; and reliance on MotionCLIP limits adaptability to new motion descriptions. Addressing these challenges calls for improved adaptation strategies and more advanced motion-language models.

## Acknowledgments

This research was funded in part by a Google South Asia & Southeast Asia Research Award given to A. Yao.

## Impact Statement

This paper presents work whose goal is to advance the field of Machine Learning. There are many potential societal consequences of our work, none of which we feel must be specifically highlighted here.

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

Table 8: Comparison with different EMA hyperparameter $\alpha$ on the 3DPW dataset. $\alpha = 0.9$ yields best performance, balancing estimated 2D pose and adapted model feedback.

| $\alpha$ | 0.75 | 0.80 | 0.85 | 0.90 | 0.95 |
|---|---|---|---|---|---|
| MPJPE | 78.6 | 77.9 | 76.8 | **76.4** | 77.5 |

This appendix includes four sections: **A.** EMA Hyperparameter, **B.** Text Labeling, **C.** More Qualitative Results, as referred in the manuscript.

## A. EMA Hyperparameter

We conduct a grid search on the exponential moving average (EMA) hyperparameter $\alpha$ to determine its optimal value. As shown in the Tab. 8, performance steadily improves as $\alpha$ increases up to 0.90, where the lowest MPJPE of 76.4mm is achieved. This indicates that most estimated 2D poses are reliable for guiding model adaptation. However, noisy estimated 2D poses still require refinement from the model prediction for better guidance (*e.g.*, at $\alpha = 0.95$).

## B. Text Labeling

In this section, we explain our text labeling process and its accuracy evaluation. Text labeling is conducted using GPT-4o (Achiam et al., 2023) as the VLM model. The evaluation is carried out through a CLIP (Radford et al., 2021) consistency check and manual correction.

As MotionCLIP (Tevet et al., 2022) has limited generalization on unseen text, GPT-4o is required to retrieve the text label from the motion dictionary used in (Tevet et al., 2022). We use the following prompt to ensure GPT-4o interprets the dictionary accurately:

> Hi, the attached dictionary motion_dict.csv contains text labels of various human motions. Please help confirm your understanding of the dictionary by answering the following:
>
> - How many labels are related to walking?
>
> - Identify the most relevant text label from the dictionary for a description like: "a person lying on the ground, with limbs outstretched".

We can obtain the following output:

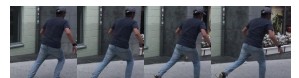 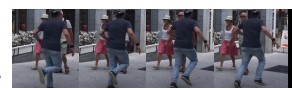

Figure 6: A video segment example used as input for text labeling.

> Here are the results based on the uploaded motion dictionary:
>
> - The number of labels related to walking: 191.
>
> - Most relevant text label: lying stretch.
>
> Let me know if you need further analysis or clarification!

By modifying "walking" and motion descriptions, we will repeat four additional similar processes to verify GPT-4o's ability to *retrieve the text label relevant to a description*. Specifically, we modify "walking" to "stand up", "talking", "go up stairs" and "pick up", which results in 72, 13, 18, and 67 identified labels, respectively. The identified labels include the motion string, or its interchangeable variations, such as replacing "go up" with "walk up" or "talking" with "argue".

More descriptions provided by us and the retrieved text labels by GPT-4o are as follows:

- a person bending down to the ground, using both hands to pick up an object carefully.

    ○ picking up something using both hand.

- a person drinking from a bottle, head tilted back, then lowering the bottle and placing it gently on the table.

    ○ drink from bottle and put down bottle.

- a person rising to a standing position, then lowering into a squat with knees bent and arms extended forward.

    ○ rise up and squat.

- a person hopping forward on the left foot, with the right leg bent slightly and arms raised for balance.

    ○ hopping on left foot.

We observe from the experiment results that GPT-4o can identify semantically similar labels and match text descriptions with an appropriate text label. We then use the following prompts to create a formatted video segment label:

You are a reliable assistant who will follow ALL rules and directions entirely and precisely. I need your assistance in creating a text label that describes the motion in a video segment. You need to first generate descriptions for the video segment. Then, you need to assign the most appropriate text label for the segment using the motion dictionary provided in the motion_dict.csv file. Below are the three examples of your answers:

- Video segment description: The person in blue jeans remained seated and talked to the person next to him throughout the video segment.
  Text label: sitting.

- Video segment description: The person in the video throws his hands upwards as if throwing something, while also walking forward.
  Text label: throw something with the right hand and walking.

- Video segment description: The person is performing figure skating on the ice, specifically executing a double Axel jump.
  Text label: None.

Your answers must be grounded in the provided video segment and follow this format. Limit segment descriptions to a maximum of 20 words. The text label must be selected from the dictionary; if no suitable label exists, return "None" as the text label.

---

- Video segment description: The person in blue jeans is running forward while adjusting his posture and avoiding obstacles.

- Text label: running.

---

For each video, we first downsample both images and sequences. Then, we divide the videos into 60-frame segments, which are subsequently fed as input to GPT-4o along with the above prompts. Below is an example of output for a video segment as shown in Fig. 6.

For evaluation, our goal is to verify whether the retrieved text label correctly reflects the human motion depicted in the video segment. We utilize CLIP (Radford et al., 2021) to measure the alignment between the text label and video segment by calculating the cosine similarity between their respective feature representations and averaging over the segment. Empirically, we observe that a text label aligning well with video semantics usually has a cosine similarity greater than 0.3, as shown in Fig. 7. For cases where the similarity is lower than this threshold, manual correction

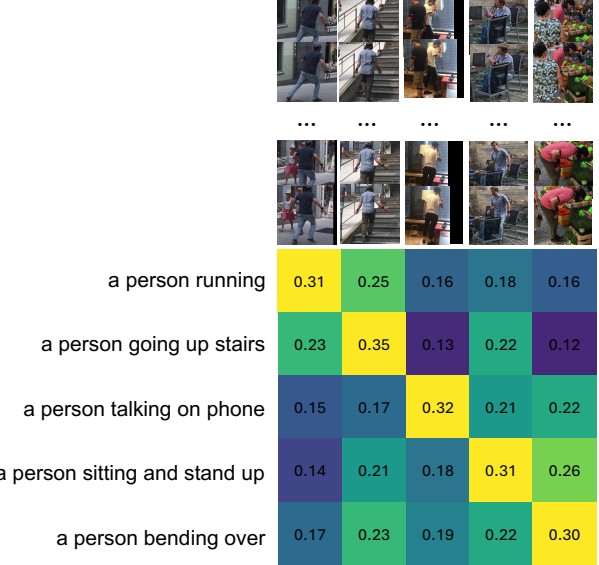

Figure 7: The cosine similarity between video segments and text labels from 3DPW dataset (Von Marcard et al., 2018). Higher similarity indicates a closer semantic alignment between text labels and video segments. This result shows that the cosine similarity of CLIP can help label verification.

is applied. Ultimately, we ensure that all video segments achieve a high alignment between text labels and video semantics, with an average cosine similarity of 0.32.

## C. More Qualitative Results

In this section, we provide more qualitative results on 3DPW (Von Marcard et al., 2018), 3DHP (Mehta et al., 2017) and EgoBody (Zhang et al., 2022) datasets. As the 3DHP dataset lacks ground truth SMPL parameters, the qualitative results on 3DHP are limited to comparisons with other methods. In 3DPW (shown in Fig. 8), our method can provide more semantics-aligned predictions for cases where all 2D are visible (running case) or some 2D are invisible (walking case). In 3DHP (shown in Fig. 9), our method captures motion details more accurately in local poses, including hand positions, leg configurations, and arm movements. On the more challenging EgoBody dataset (shown in Fig. 10) with severe body truncation, our method excels in aligning with motion semantics.

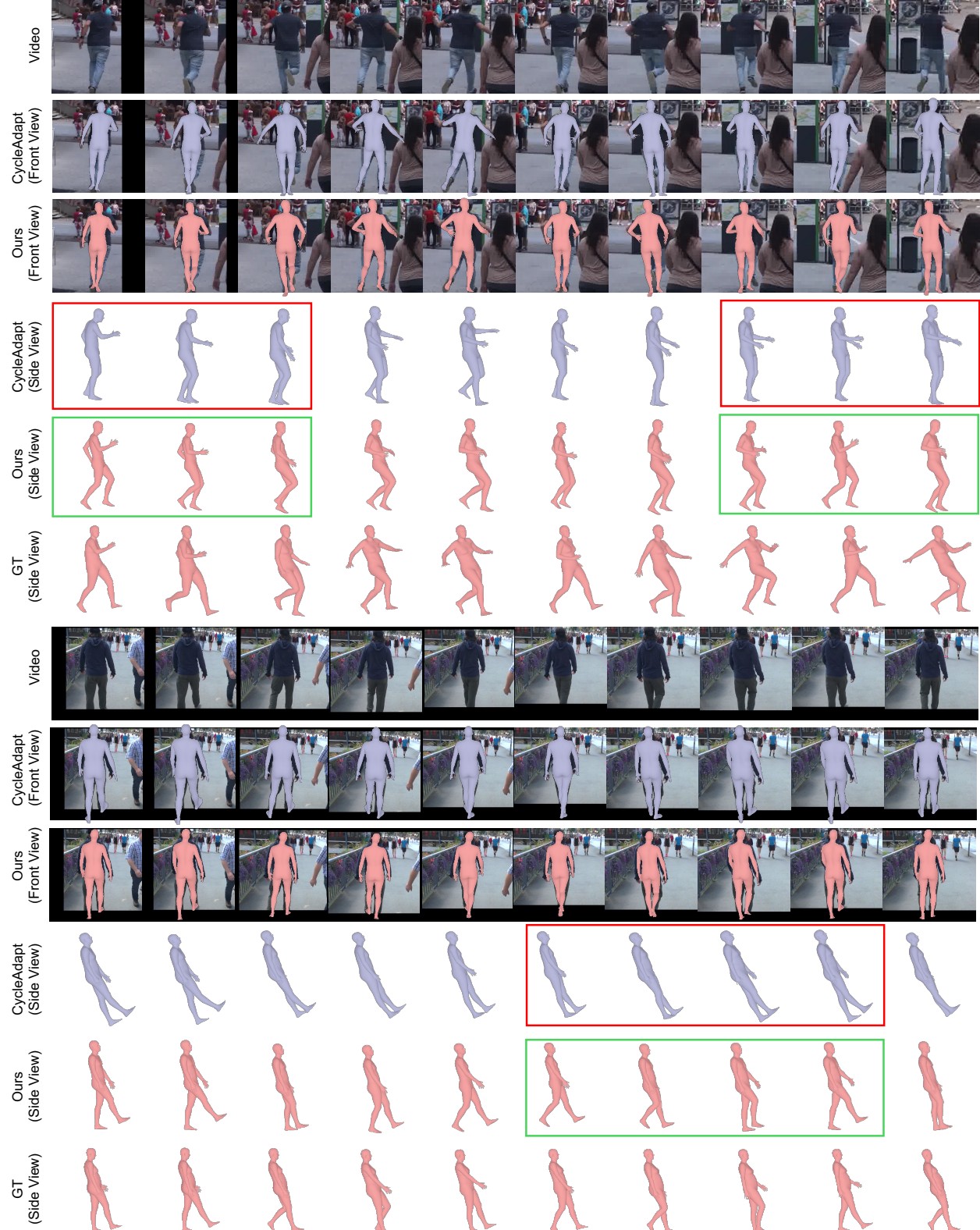

Figure 8: Qualitative comparison on two video segments from 3DPW dataset (Von Marcard et al., 2018). Our method demonstrates more semantics-aligned predictions for the "**Running**" and "**Walking**" motions. The red and green squares highlight the examples where CycleAdapt fails, but our method performs better.

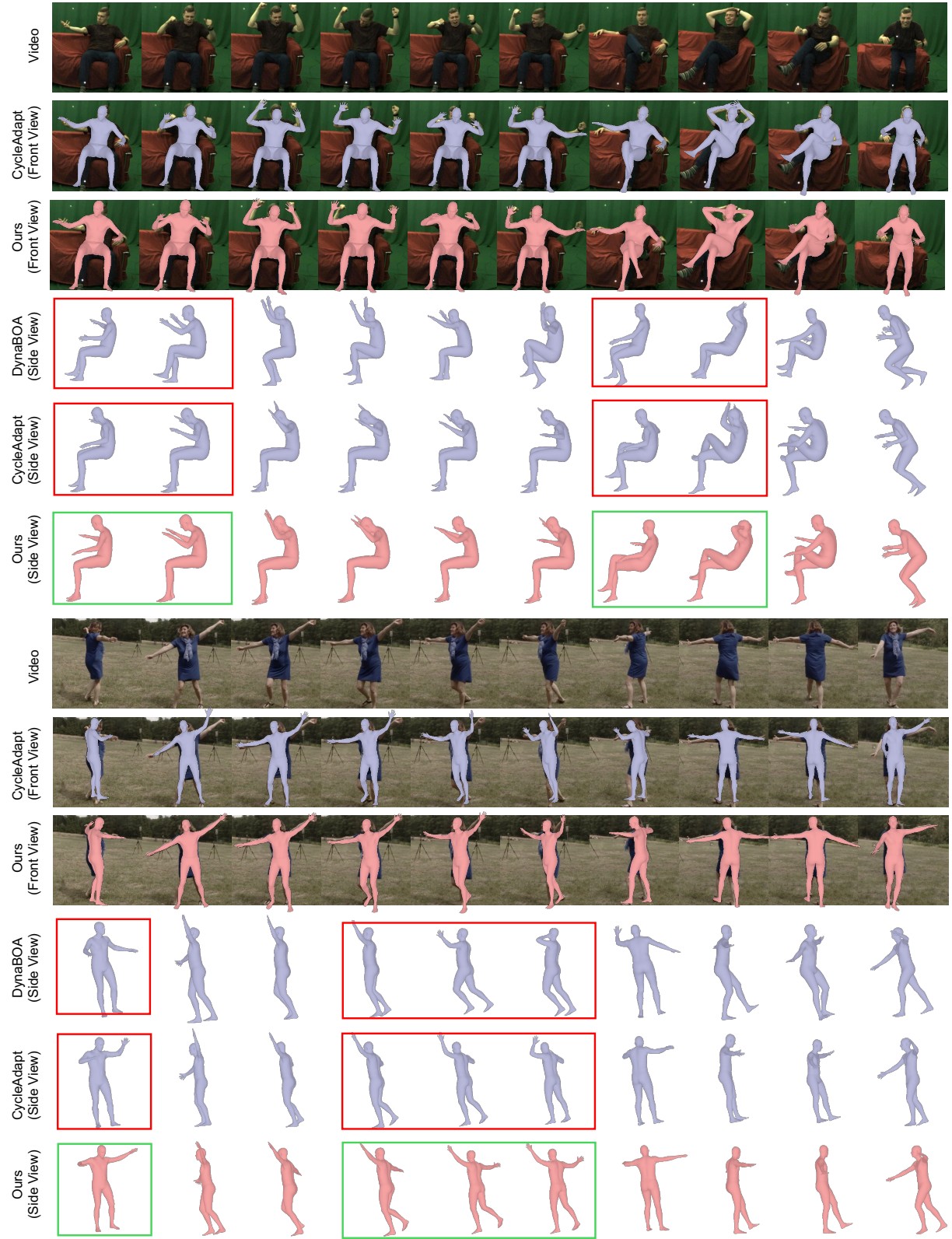

Figure 9: Qualitative comparison on two video segments from 3DHP dataset (Mehta et al., 2017) with text labels of "**Sitting**" and "**Spin around with right foot**". Our method provides more accurate pose predictions in the motion details compared to CycleAdapt (Nam et al., 2023) and DynaBOA (Guan et al., 2022). For example, we can better capture hand positions while sitting (1st segment), crossing legs while sitting (1st segment), and arm movements while spinning around (2nd segment).

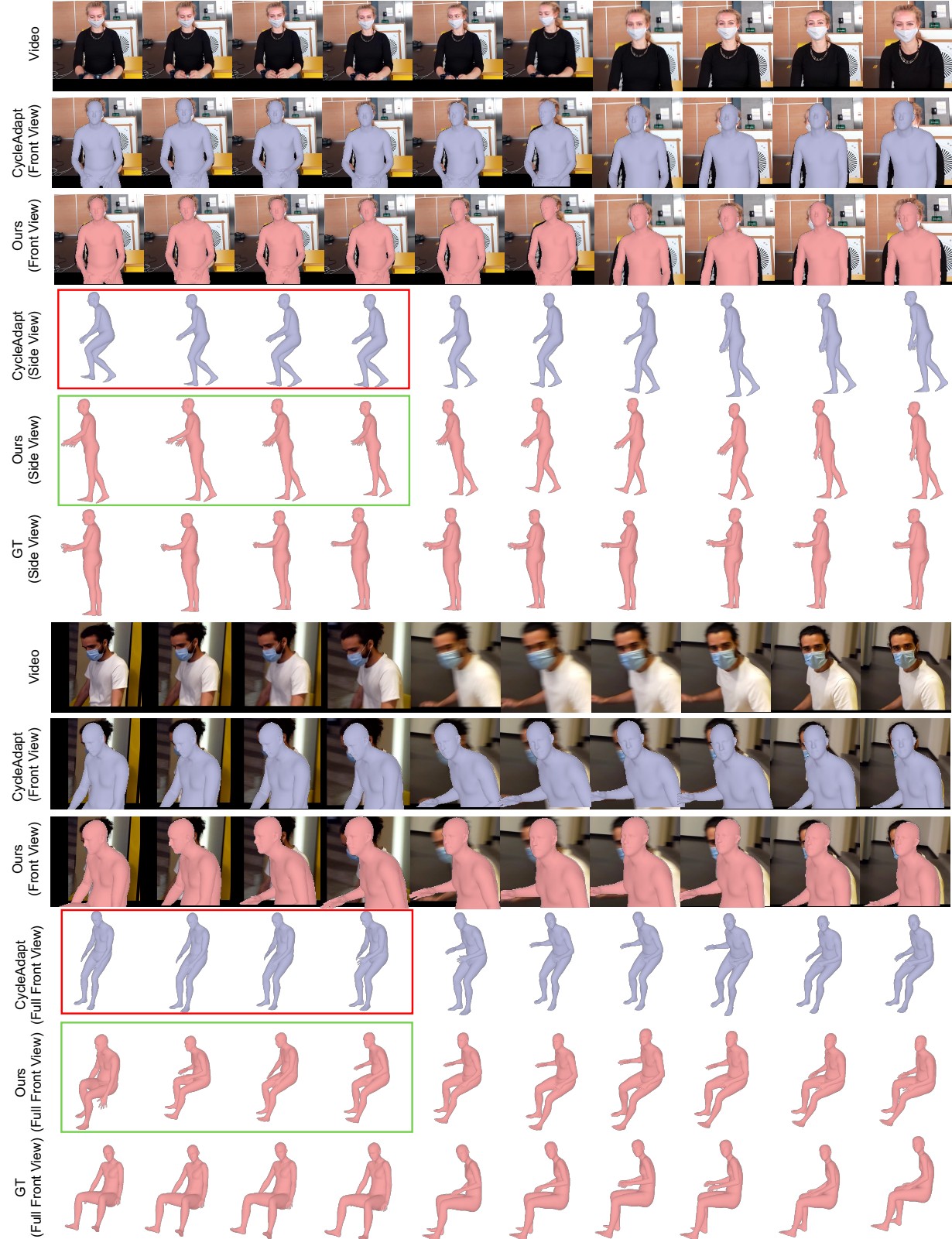

Figure 10: Qualitative comparison on two video segments from EgoBody dataset (Zhang et al., 2022).We can enhance the pose predictions to be aligned with "**Standing**" and "**Sitting**" text labels. The red and green squares highlight the examples where CycleAdapt fails, but our method performs better.

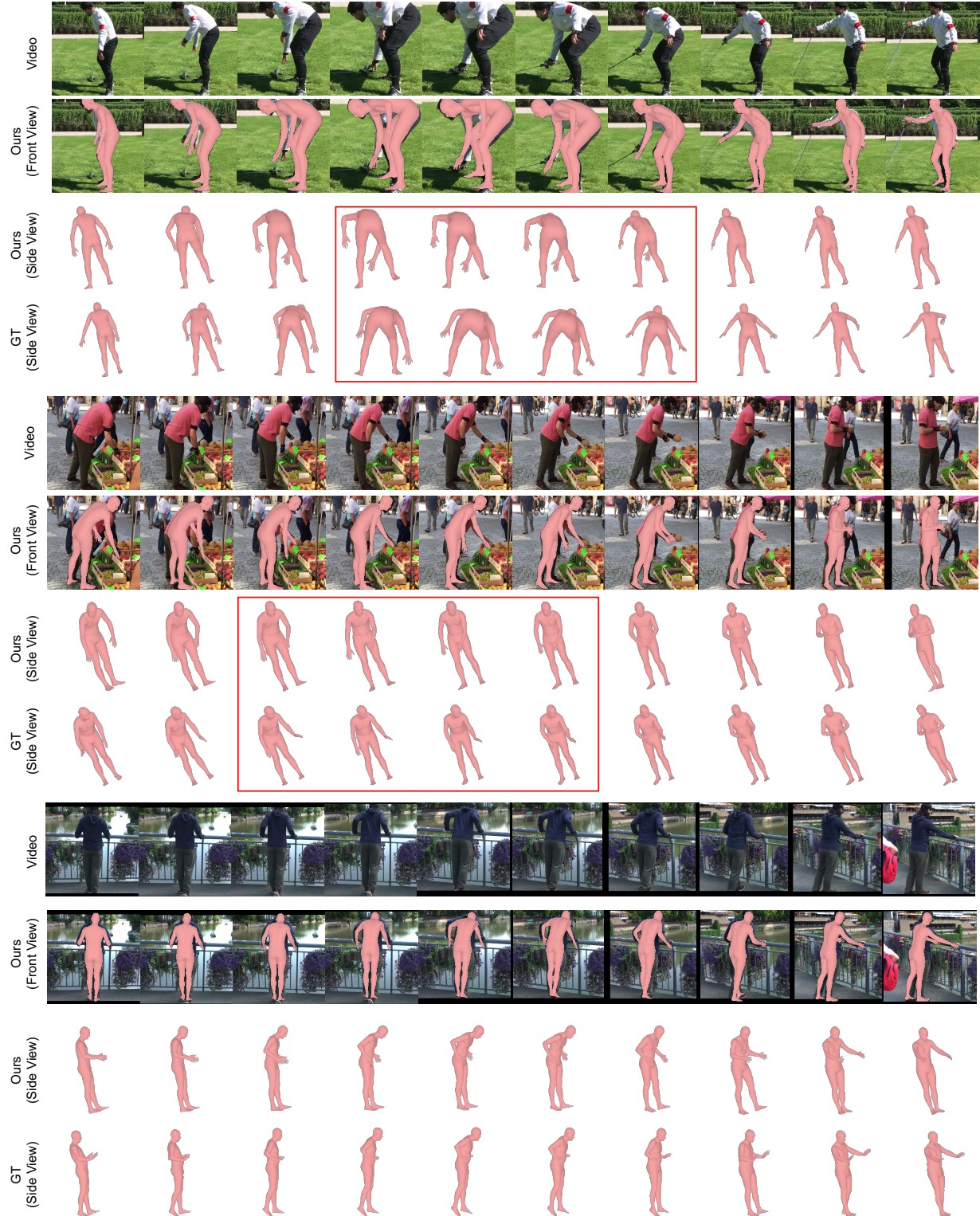

Figure 11: VLM failure cases: "**golfing**", "**eating something using both hands**", and "**cast fishing pole**". Despite VLM hallucination causing some text labels to misalign with action semantics, our method remains robust. Additionally, some VLM predictions and GT exhibit different motion directions (*e.g.*, depth ambiguity of picking up from the front vs. right front in the first example), which may be interesting to consider in future work.

