# OpenReview forum: "Semantics-aware Test-time Adaptation for 3D Human Pose Estimation"
_ICML.cc/2025/Conference — ICML 2025 poster_

### Official Review · Reviewer_qh14 · 2025-03-11

**Overall Recommendation:** 3

**Summary:**

This paper presents a 3D human pose estimation that supports test-time optimization with semantics. The authors leverage video understanding and a well-structured motion-text space to adapt the model motion predictions. In addition, they incorporate a missing 2D pose completion with the motion-text similarity. Experimental results demonstrate the effectiveness of the proposed method.

## update after rebuttal

The authors address my questions well. I keep my original rating.

**Claims And Evidence:**

yes

**Essential References Not Discussed:**

no

**Experimental Designs Or Analyses:**

yes

**Methods And Evaluation Criteria:**

yes

**Other Comments Or Suggestions:**

Please see weaknesses.

**Other Strengths And Weaknesses:**

Strengths

As the authors argued, existing TTA methods lack semantics, which can suffer from many failure cases, including the top row of Figure 1. Adding semantics with a vision-language model is a good step toward semantics-aware 3D human pose estimation. The effectiveness of such incorporation is well-shown in the experiments section.

Weaknesses
1. It is not clear how robust the system is to the wrong outputs from VLM. It could be interesting if the authors could show some failure cases of the proposed system including the failure cases from the VLM.
2. The results in Figure 1 below are not appealing. Due to the occlusion, it is not clear what is the correct one between CycleAdapt and the proposed method’s outputs.
3. The running time should be much slower than CycleAdapt due to the VLM. What is the exact running time in the TTA?.
4. Overall, qualitative results only include easy poses, such as simply walking or standing. More challenging poses should be included.

**Questions For Authors:**

1. What if we directly optimize the output 3D poses without fine-tuning the 3D HPE model? I think that should make the TTA much faster.

**Relation To Broader Scientific Literature:**

It is related to many human-centric computer vision tasks.

**Theoretical Claims:**

yes

---

> ### Author Rebuttal · Authors · 2025-04-01
>
> We express our sincere appreciation for the helpful reviews and tackle the concerns below:
>
> **[Weakness 1] Failure cases from the VLM**
>
> **R:** Thank you for your suggestions. We manually examined 5,000 frames from the 3DPW dataset, where 96.4% is accurate, as the actions are relatively simple and VLM is reliable. For the remaining 3.6%, the VLM hallucinates actions based on background, e.g. a person picking up foil is mistaken as "golfing" as the foil resembles a golf club, or "eating something using both hands" and "cast fishing pole" in scenes with fruit stands and rivers, respectively. Visualizations of these VLM's failure cases can be found in Fig.I ([anonymous GitHub link](https://anonymous.4open.science/r/visualization-D59C/anonymous_pdf.pdf)).
>
> Despite these VLM failures, our method maintains robustness. As our semantics-aware alignment is a regularizer, its impact is limited if it conflicts with 2D projection loss.  The table below shows that the adaptation with the wrong VLM label still reduces the MPJPE, but not as much as when the (oracle) correct action labels were applied.
>
> | Method | Initial | Incorrect VLM action output | Correct (oracle) action output |
> | --- | :---: | :---: | :---: |
> | MPJPE (mm) 	| 213.8 | 92.2 | 83.7 |
>
>
>
> **[Weakness 2] No GT mesh in Figure 1**
>
> **R:**  Thanks for pointing this out. As shown in Fig.IV ([anonymous GitHub link](https://anonymous.4open.science/r/visualization-D59C/anonymous_pdf.pdf)), we include GT mesh for a better comparison, where our predictions preserve the same motion semantics as the GT.
>
>
> **[Weakness 3] Runtime analysis**
>
> **R:** Thanks for pointing this out.  Yes, our runtime (107.6ms) is longer than CycleAdapt (74.1ms); the extra overhead is primarily from the VLM (85.4%).
>
> Details are given in our response to **[Question 1] (Runtime analysis)** for the reviewer **qP6M**.
>
> **[Weakness 4] Include challenging poses**
>
> **R:** Thank you for your suggestions. We provide more challenging poses, i.e., "open a door and sit", "hugging", "squats series" and "spin around with right foot". Please refer to Fig.II and Fig.III in the [anonymous GitHub link](https://anonymous.4open.science/r/visualization-D59C/anonymous_pdf.pdf).
>
>
> **[Question 1] Direct optimization without fine-tuning**
>
> **R:**  Yes, it is faster since it does not require model fine-tuning. However, the success of direct optimization is linked heavily to the accuracy of the initial estimate.
>
> We compare with ScoreHMR [3], a state-of-the-art direct optimization approach on the 3DPW dataset using OpenPose 2D poses. As shown in Fig.V ([anonymous GitHub link](https://anonymous.4open.science/r/visualization-D59C/anonymous_pdf.pdf)), our method is robust across different initial prediction qualities. ScoreHMR is more sensitive and when the initial MPJPE exceeds 200mm, ScoreHMR reduces error only by 26.6%, whereas we can reduce it by 64.5%.
>
> **References**
>
> [3] Stathopoulos et al. Score-guided diffusion for 3D human recovery. CVPR'2024.

---

### Official Review · Reviewer_qP6M · 2025-03-13

**Overall Recommendation:** 4

**Summary:**

The paper introduces a TTA HMR method from videos using semantic information to address challenges caused by a lack of information when a large portion of the body is occluded. The paper suggests using MotionCLIP to align predicted motions with the CLIP representation of the textual description of the actions as extracted by GPT-4o. The 3D HPE model is then adapted during test time to match OpenPose keypoints and achieve semantic alignment while being temporally regularized. Next, the fine-tuned HMR model obtains initial predictions, which are further improved using a semantic-aware EMA operation. The main finding is that incorporating this semantics-aware motion prior significantly improves performance over state-of-the-art TTA methods, especially in cases with occlusions or truncations.

**Claims And Evidence:**

The paper claims improved performance over SOTA, which is supported in the experiments by showing the effectiveness of adding semantic information. The paper also propose an interesting approach to incorporate the new modality and through ablations, shows the effectiveness of fill-ins and EMA.

**Essential References Not Discussed:**

The paper understands related literature well, citing adequate and recent references from the last two years. The model lacks a reference to diffusion models, like RoHM and MDM, which also use trajectory information. However, this lack of reference is acceptable since that line of research focuses on motion generation.

**Experimental Designs Or Analyses:**

The experimental design and experiments seem sound. The paper claims that it follows the same evaluation protocol as prior research and uses recent papers for comparison.

**Methods And Evaluation Criteria:**

The datasets and evaluation metrics are commonly used in this field. Using multiple datasets and ablations is appropriate to show the effectiveness of the proposed method. The experiments might be unfair in Table 1, since none of the other methods use semantic information, but it's acceptable given the context of the paper.

**Other Comments Or Suggestions:**

I did not find any major issues. Please address my questions below if possible. Thank you.

Typo:
- L302, right column, "Egobody" should be "EgoBody"

**Other Strengths And Weaknesses:**

### Strengths
1. The paper is well-written, clear, and concise. It is easy to follow and provides sufficient details to understand the paper. Although some components are too reliant on the prior works, the paper has done a good job summarizing their efforts.
2. The supplementary materials provides adequate information to replicate the experiments.
3. The paper has a good understanding of the literature, citing and comparing with them when needed.
4. The paper provides a compelling approach to incorporate sematic information into TTA, improving upon previous works.

### Weaknesses
1. The paper does not provide any additional alternatives other than previous works.
2. As the authors mentioned, this method cannot perfectly capture the correct movements of the subjects while they are occluded. However, it could generate a plausible motion based on semantic consistency. There is no information about long periods of occlusion (whole body) or jitters (due to thresholding).

**Questions For Authors:**

1. Could you please provide a runtime analysis, like Table 7 of the CycleAdapt paper? It is crucial to understand the computational requirements of training/testing of your method compared to others.
2. How did you choose the hyperparameters? Wouldn't the EMA and $\alpha$ cause a jitter in motion during different actions?
3. Could you provide the results of using GT 2D human keypoints instead of OpenPose for table 1? It would be interesting to understand the noise robustness of your approach compared to others when working with GT/OpenPose.

**Relation To Broader Scientific Literature:**

Motion priors trained on large motion capture datasets like AMASS have been used to regularize 3D human pose and motion estimation, with early examples like VPoser for pose regularization and later works like HuMoR employing VAEs for motion priors. Recent Test-Time Adaptation (TTA) approaches have demonstrated that fine-tuning 3D Human Mesh Recovery (HMR) models on test video sequences improves speed and accuracy. This paper builds upon TTA, proposing incorporating semantic information via a pre-trained motion-language model into fine-tuning. While the paper leverages existing components from the literature, it presents a well-justified strategy for addressing occlusions through semantic consistency during TTA.

**Theoretical Claims:**

The paper does not have any theoretical proof.

---

> ### Author Rebuttal · Authors · 2025-04-01
>
> We express our sincere appreciation for the helpful reviews and tackle the concerns below:
>
> **[Method 1] Fairness in Table 1**
>
> **R:** Our paper highlights the problem of motion semantics and proposes a method to incorporate semantics as the core contribution.  No existing works use semantics, so it is difficult to compare directly. We verify our method by comparing it with the closest components from the literature, as shown in Table 3. Specifically, we use CycleAdapt as a baseline (top row) and then add semantics through a motion discriminator (row 2) or unpaired local poses (row 3). In this way, we enforce the predicted motion to be aligned with the generated motion from [1] by sharing the same semantics as the video segment. Table 3 in the paper shows that our method better integrates semantics without additional motion data generation.
>
> **[References 1] Diffusion models**
>
> **R:** Thank you for your suggestion. We will add these references and also discuss diffusion models in our related works section.
>
>
> **[Weakness 1] Method alternatives**
>
> **R:** Thanks for pointing this out. We explore different strategies of integrating semantic information; two alternatives to our method are shown in Table 3. We will adjust the labels and captions of Table 3 and Sec. 5.4 (‘Analysis on semantics-incorporated techniques’) in the text to clarify this point.
>
> **[Weakness 2] Long periods of occlusion or jitters**
>
> **R:** Thank you for pointing this out.
>
> For long periods of full-body occlusion, motion can be generated to maintain semantic consistency with the predicted activity. However, this aligns more with motion generation, which is evaluated by plausibility (e.g., FID [2]), unlike reconstruction, which relies on ground truth. We plan to merge reconstruction and generation into a single framework and develop suitable evaluation metrics in future work.
>
> We evaluate jitter on the 3DPW dataset, comparing our method with ground truth and CycleAdapt. As shown in the table below, our jitter is significantly lower than the image-based DynaBOA but 11.1% higher than CycleAdapt. However, CycleAdapt often predicts static motion, even in dynamic motion scenes, resulting in low jitter but deviating from the ground truth. Take the walking motion from Fig. 1 of the paper as an example, we report the jitter values for this case in the last row of the table below, where ours is higher than CycleAdapt but closely aligns with the ground truth.
>
> | Jitter$(10m/s^3)$ | GT | CycleAdapt | DynaBOA | Ours |
> | --- | :---: | :---: | :---: | :---: |
> | All sequences | 22.3 | 27.9 | 172.1 | 31.0 |
> | Walking | 46.7 | 16.8 | - | 48.3 |
>
>
> **[Other Comments 1] Typo**
>
> **R:** Thank you for pointing this out. We will correct it in the revision.
>
> **[Question 1] Runtime analysis**
>
> **R:** Thank you for your suggestion. We perform runtime analysis following the environment settings described in Table 7 in CycleAdapt. The table below shows that compared to CycleAdapt, our method introduces an additional 33.5ms runtime per frame, 85.4% of which is caused by VLM, and 14.6% by our proposed framework components. However, it is still much faster than other TTA methods. We will add runtime analysis in the revision.
>
> | Method | BOA | DynaBOA | DAPA | CycleAdapt | Ours |
> | --- | :---: | :---: | :---: | :---: | :---: |
> | Runtime(ms)  | 840.3 | 1162.8 | 431.0 | **74.1** | 107.6 (+33.5) |
>
>
> **[Question 2] Hyperparameters and EMA**
>
> **R:** We choose hyperparameters based on a grid search. The EMA update factor ($\alpha$) values used for the search are provided in the table below. For the fill-in threshold ($\sigma$) of missing 2D detection, please refer to Table 5 in the paper.
>
> The EMA and $\alpha$ do not cause jitters. This is because the EMA is performed over 2D poses for each frame and each frame has one fixed action label.
>
> | $\alpha$| 0.75 | 0.80 | 0.85 | 0.90 | 0.95 |
> | --- | :---: | :---: | :---: | :---: | :---: |
> | MPJPE(mm)| 78.6 | 77.9 | 76.8 | **76.4** | 77.5 |
>
>
> **[Question 3] GT 2D**
>
> **R:** Thank you for your suggestion. We add results in the table below. Our method still performs better than CycleAdapt. The improvement becomes smaller because GT 2D keypoints provide very strong indications for occluded keypoints, which is unrealistic in real-world scenarios. Our method especially takes effect when 2D keypoints are noisy or missing.
>
> | Method |      |3DPW  |          | 3DHP      |          |
> | --- | :---: | :---: | :---: | :---: | :---: |
> |       |    | MPJPE     | PA-MPJPE | MPJPE     | PA-MPJPE |
> | OpenPose | CycleAdapt | 87.7     | 53.8      | 110.3    | 74.4     |
> |        | Ours      | 76.4     | 47.2      | 101.3    | 65.1     |
> | GT     | CycleAdapt | 64.7     | 39.9      | 100.9    | 64.6     |
> |        | Ours      | **64.1** | **39.4** | **98.5** | **63.7** |
>
>
> **References**
>
> [1] Jiang et al. MotionGPT: Human motion as a foreign language. NeurIPS'2023.
>
> [2] Guo et al. Generating diverse and natural 3d human motions from text. CVPR'2022.

---

> > ### Comment · Reviewer_qP6M · 2025-04-07
> >
> > I thank the authors for their comprehensive rebuttal, which has addressed all my concerns. Specifically, I appreciate the detailed runtime analysis confirming the method's efficiency relative to other approaches. Furthermore, the analysis using Ground Truth 2D keypoints highlights the method's strength in handling the noisy or missing detections typical in real-world scenarios. The clarifications regarding EMA and the quantitative jitter analysis were also convincing, fully resolving my reservations on those aspects. Therefore, I maintain my score and recommend acceptance of this paper.

---

> > > ### Author Response · Authors · 2025-04-08
> > >
> > > Thank you for recognizing our responses in the rebuttal! We are glad that the clarifications and analyses addressed your concerns. We truly appreciate your recommendation for acceptance and will incorporate your valuable feedback into the revision.

---

### Official Review · Reviewer_c9x6 · 2025-03-13

**Overall Recommendation:** 4

**Summary:**

This paper proposes a novel semantics-aware test-time adaptation (TTA) framework for 3D human pose estimation, addressing the issue of overly smoothed or unguided predictions, especially under occlusion or truncation. The key innovation is integrating motion semantics into the TTA process by leveraging MotionCLIP to align pose sequences with textual action labels inferred via a vision-language model (VLM). Additionally, missing 2D keypoints are completed based on motion-text similarity to improve supervision where 2D evidence is lacking. The method significantly outperforms prior approaches such as CycleAdapt on datasets including 3DPW, 3DHP, and EgoBody. Supplementary results show high semantic consistency, effective fill-in mechanisms, and robust improvements in both common and rare motion types.

**Claims And Evidence:**

The authors' claims are well supported:
1. The semantics-aware motion prior reduces depth ambiguity and guides the adaptation process.
2. 2D pose completion using text-aligned motion enhances adaptation under occlusion.

**Essential References Not Discussed:**

N.A.

**Experimental Designs Or Analyses:**

Strong design and analysis. The paper includes:

1. Solid baselines (BOA, DynaBOA, DAPA, CycleAdapt).
2. Ablation studies isolating the contribution of alignment, EMA, and fill-in components.
3. Per-semantic motion improvement analysis.
4. Supplementary evaluations on fill-in threshold impact and text-labeling accuracy.

**Methods And Evaluation Criteria:**

The methods are carefully designed and appropriate:

1. MotionCLIP is used effectively as a semantics-aware regularizer.
2. A VLM (GPT-4o) assigns action labels to video segments, with verification via CLIP cosine similarity.
3. EMA and fill-in strategies help refine 2D poses over adaptation epochs.

The evaluation uses standard benchmarks (3DPW, 3DHP, EgoBody) and metrics (MPJPE, PA-MPJPE, MPVPE).

**Other Comments Or Suggestions:**

N.A.

**Other Strengths And Weaknesses:**

N.A.

**Questions For Authors:**

N.A.

**Relation To Broader Scientific Literature:**

This paper advances the state-of-the-art in semantics-aware adaptation for 3D human pose estimation, bridging two previously disjoint areas: vision-language modeling and test-time pose refinement. While MotionCLIP and CycleAdapt are relevant foundations, this work introduces a novel integration of motion semantics during adaptation. It complements prior efforts in temporal smoothing and motion priors by adding a high-level semantic guidance layer.

**Theoretical Claims:**

There are no deep theoretical claims in this work.

---

> ### Author Rebuttal · Authors · 2025-04-01
>
> We express our sincere appreciation for the helpful reviews and welcome further discussion!

---

> > ### Comment · Reviewer_c9x6 · 2025-04-08
> >
> > There are no further issues from my side. I will keep my score at 4: Accept.

---

> > > ### Author Response · Authors · 2025-04-08
> > >
> > > Thank you for your time and effort in reviewing our paper! We truly appreciate your initial positive score and your decision to maintain it.

---

### Decision · Program_Chairs · 2025-05-01

**Decision:**

Accept (poster)

**Comment:**

This paper achieved unanimously positive reviews among reviewers and has made solid contributions in 3D human pose estimation. The initial concerns are mostly about evaluation and the authors have addressed them well during rebuttal.

Therefore, I recommend acceptance of this paper and encourage the authors to integrate the rebuttal in their revision.